# Prevalence and factors associated with anogenital warts among sexual and gender minorities attending a trusted community health center in Lagos, Nigeria

**Sylvia B. Adebajo**[1]◉*, **Rebecca G. Nowak**[2]◉, **Ruxton Adebiyi**[1], **Elizabeth Shoyemi**[3], **Charles Ekeh**[3], **Habib O. Ramadhani**[2], **Charlotte A. Gaydos**[4], **Julie A. Ake**[5], **Stefan D. Baral**[6], **Manhattan E. Charurat**[2], **Trevor A. Crowell**[5,7]◉, for the TRUST/RV368 Study Group¶

1 Center for International Health Education Biosecurity, University of Maryland, Baltimore, Abuja, Nigeria,
2 Institute of Human Virology, University of Maryland, Baltimore, Maryland, United States of America,
3 Population Council Nigeria, Abuja, Nigeria, 4 Department of Medicine, Johns Hopkins University, Baltimore, Maryland, United States of America, 5 United States. Military HIV Research Program, Walter Reed Army Institute of Research, Silver Spring, Maryland, United States of America, 6 Johns Hopkins Bloomberg School of Public Health, Baltimore, Maryland, United States of America, 7 Henry Martin Jackson Foundation for the Advancement of Military Medicine, Bethesda, Maryland, United States of America

◉ These authors contributed equally to this work.
¶ Membership of the TRUST/RV368 Study Group is listed in the Acknowledgments.
* SAdebajo@mgic.umaryland.edu

**Data Availability Statement:** The dataset underlying the findings reported in this manuscript

## Abstract

Anogenital warts caused by human papillomavirus are common in sexual and gender minorities (SGM). The prevalence of, and factors associated with warts were described for SGM with a high burden of HIV in Nigeria. Individuals who reported anal sex with men were enrolled in the TRUST/RV368 cohort. Participants completed an interviewer-led survey, provided biological samples, and had a physical examination. Specific to the Lagos site, clinic staff offered standardized warts treatment services. RDS-weighted multivariable logistic regression was used to estimate the adjusted odds ratios (AORs) and 95% confidence intervals (CIs) for factors potentially associated with anogenital warts. Of 672 enrolled SGM, 478 (71%) engaged in warts services and had complete data. The median age (interquartile range) was 22 (20–26) years, 272 (52%) initiated sex before age 18, and 347 (79%) were cisgender men. Multiple male sexual partners in the previous year were reported by 448 (90%) of the participants, and 342 (66%) were living with HIV. Warts were diagnosed in 252 (54%), including anal warts in 234 (43%) and penile warts in 44 (8%); 26 (5%) had both anal and penile warts. Factors independently associated with warts included HIV (AOR:2.97; CI:1.44–6.14), engaging in receptive anal sex (AOR:3.49; CI:1.25–9.75), having multiple male sexual partners (AOR:7.26; CI:2.11–24.87), age at sexual debut (AOR:0.53; CI:0.28–0.98), and non-binary gender identity (AOR:0.20; CI:0.05–0.71). Warts were common among SGM in Nigeria, particularly those living with HIV. Administration of HPV vaccination before sexual debut or as a catch-up vaccination may prevent HPV-associated complications.

is available and can be accessed from Dryad: https://datadryad.org/stash/share/_LltoWZQoZ AH1QWrqppfA8Vzt48LbiCKL8Fc4iXhHl4 If the file can't be accessed, please press reload for the documents to be downloaded. Thanks.

**Funding:** This study was supported by cooperative agreements between the Henry M. Jackson Foundation for the Advancement of Military Medicine, Inc., and the U.S. Department of Defense [W81XWH-11-2-0174, W81XWH-18-2-0040], the National Institutes of Health [R01 MH099001, R01 AI120913, R01 MH110358, K07CA225403, R01HL165686], Fogarty Epidemiology Research Training for Public Health Impact in Nigeria program [D43TW010051], and the President's Emergency Plan for AIDS Relief through a cooperative agreement between the Department of Health and Human Services/Centers for Disease Control and Prevention, Global AIDS Program, and the Institute for Human Virology-Nigeria [NU2GGH002099]. The U.S. Army was one of several entities that provided funding for this research. U.S. Army investigators were involved in the study design, study operations, data collection, data analysis, data interpretation, and writing of the report.

**Competing interests:** The authors have declared that no competing interests exist.

## Introduction

Anogenital warts, also known as condylomata acuminata, are the most common clinical manifestation of human papillomavirus (HPV) infection. They are highly infectious, recurrent, expensive to treat, distressing, and stigmatizing sexually transmitted infections (STI) that impact negatively on the quality of life of men and women globally [1–4]. The worldwide prevalence of anogenital warts is estimated at 0.13%-5.1% with an incidence of 103–170 cases per 100,000 person-years among men and 76–190 per 100,000 person-years among women [1, 4].

HPV comprises a diverse group of viruses with over 100 genotypes, most of which manifest as asymptomatic infections that resolve without treatment. Genotypes classified as low-risk (LR-HPV) and high-risk (HR-HPV) cause persistent non-oncogenic and oncogenic clinical diseases, respectively [5]. Approximately 90% of all anogenital warts are caused by LR-HPV genotypes 6 and 11 [6], while 80–90% of anogenital cancers are causally linked to HR-HPV genotypes 16 and 18 [5]. HPV 6 and 11 have also been found as the only genotypes in some anal cancers [7]. Furthermore, coinfections of anogenital LR-HPV and HR-HPV genotypes are common among sexual gender minorities (SGM) [8].

Globally, SGM and, in particular, those living with HIV are at a higher risk of LR-HPV and HR-HPV-related infections and neoplastic lesions than men who have sex exclusively with women [9]. A synergistic association between anogenital HPV and HIV has been reported; HPV infection increases the risk of HIV acquisition and HIV infection, increasing the risk of anal warts and HR-HPV-related squamous cell cancer of the anus [10, 11].

Across sub-Saharan Africa, there is mounting evidence of an increasing burden of diseases associated with HPV infection in men, with a higher prevalence of HPV-related diseases reported than in high-income countries [12]. Prevalence of anogenital warts among men ranges from 2.0% - 12.2% across the East, Central, South and West African regions [13–15] with the highest reported in West Africa [14, 16]. The incidence of anogenital warts varies from 1.4–5.3/ 100 person-years [13, 14]. Prevalent and incident anogenital wart lesions are higher in African men living with HIV. They experience more florid and prolonged clinical manifestations of anogenital warts than men not living with HIV [13, 17]. Furthermore, men living with HIV are at a higher risk of coinfections with HR-HPV, leading to high-grade squamous intraepithelial lesions (HSIL) and anal cancer [12, 13, 16]. The impact of antiretroviral therapy (ART) on the incidence and course of anogenital warts remains unclear. According to a systematic review, some studies reported that a longer duration of ART was associated with decreased incidence or persistence of anogenital warts, while others reported the converse [14]. Moreover, most of these studies were conducted among women, making it difficult to generalize the results.

A few studies have documented the burden of anogenital warts among SGM in countries across sub-Saharan Africa, especially among those living with HIV [11, 13, 15]. In Kenya, the prevalence of anogenital warts among high-risk heterosexual, homosexual, and bisexual men was 2.1%, 2.3%, and 3.4%, respectively. Moreover, the incidence of anogenital warts among SGM in Kenya was 5.3 per 100 person-years [13], almost four times higher than an earlier estimate among men employed by a trucking company in Mombasa, Kenya (1.4 per 100 person-years) [18]. In another study of 154 young SGM in Abuja, Nigeria, prevalence of anal warts was 11.8%, higher among SGM living with HIV compared to those without HIV (18.0% vs. 3.1%, p<0.001) [15].

This study aimed to estimate the prevalence and risk factors of anogenital warts in a large cohort of SGM in Lagos, Nigeria.

## Methods

This was a cross-sectional study of SGM recruited between March 2013 and March 2017 for HIV treatment and care services using respondent-driven sampling (RDS) [19] from a friendly

community health and research center as part of the TRUST/RV368 cohort study, as previously described [15, 20]. Specifically, five participants were non-randomly recruited based on the size and diversity of their social networks to form the seeds for the recruitment of subsequent SGM participants. The seeds, which formed the recruitment wave 0, were each given three coupons to recruit three peers from their social networks that created the second recruitment wave 1. The chain-referral continued through a series of recruitment waves until equilibrium was achieved when the characteristics of the recruited cohort were independent of the seeds. To be eligible, each participant had to be assigned male sex at birth, aged 18 years or older, present a valid RDS coupon, and report receptive or insertive anal intercourse with a male partner at least once during the year before enrollment.

At enrollment, each participant was administered a standardized questionnaire that elicited demographic, clinical, and behavioral data. Among the behavioral questions, participants were asked to report whether they engaged in transactional sex and their sexual position (insertive sex, receptive sex, or both) with men during the preceding 12 months. Physical examinations, including penile and anal inspections, were performed. Participants were counseled and tested for HIV using a parallel algorithm of two rapid tests with Determine (Alere, Watham, MA, USA) and Uni-gold (Trinity Biotech, Wicklow, Ireland) kits.

These analyses were restricted to the enrollment visits of participants at the Lagos site, where clinicians offered standardized warts treatment services between May 2014 and September 2016. The services included rectal inspection to diagnose anogenital warts and cryotherapy provided on designated clinic days. Depending on the size of the warts, the participants were either treated in the clinic with liquid nitrogen cryotherapy or referred to the Nigerian Army Reference Hospital 68 Yaba, Lagos, for surgical excision.

Only those who engaged in the warts treatment service and completed a behavioral survey were included in the cross-sectional analyses. Our study was approved by the Health Research Ethics Committee, FCT, Nigeria; Population Council, New York, USA; Walter Reed Army Institute of Research, Silver Spring, Maryland, USA; and University of Maryland, Baltimore, Maryland, USA institutional review boards. All participants provided written informed consent prior to enrolment.

## Outcome variable

The primary outcome of these analyses was the presence of penile or anal warts. Anogenital warts were categorized as present if the participants had anal or penile warts.

## Independent variables

Sexual orientation was categorized as homosexual, bisexual, heterosexual, or queer; gender identity was categorized as cisgender, transgender and non-binary based on participants' self-reported responses. Sexual position with male partners was categorized as: 'insertive sex only', 'receptive sex only', or 'both insertive and receptive sex'. Sexual positions with female partners were assessed and categorized separately. Condom use was defined as always used condoms or any condomless sex during insertive and/or receptive sex with both male and female partners in the preceding 12 months. In these analyses, multiple sexual partners were defined as having more than one male or female partner in the preceding 12 months. Transactional sex identified participants who did or did not exchange money and/or gifts for sex in the preceding 12 months. The use of and ease of access to lubricants during sex were also assessed, as well as HIV, ART use, viral load, CD4 count, age, age at sexual debut, marital status, education, occupation, and currently a student and recruitment waves categorized by fives (0–4,5–9,10–14,15–19,20–24,25–29) [19].

## Statistical analyses

RDS weighting was used in our analyses based on individual social network size and the number of coupons distributed. Weighted Pearson's chi-squared tests were used to compare the differences in the prevalence of anogenital warts across various demographic, biological, and behavioral characteristics. Bivariable logistic regression analyses were used to estimate the associations between anogenital warts and HIV, ART use, viral load, as well as a range of sexual behaviors, including transactional sex, sexual positioning and condom use. RDS-weighted multivariable logistic regression was used to identify factors independently associated with anogenital warts. Variables underwent backward stepwise selection for the multivariable model using a p-value threshold of 0.157, and the final multivariable model was selected based on the likelihood ratio test, collinearity consideration, and change-in-estimate criterion [21]. P-value <0.05 was considered statistically significant and adjusted odds ratios (AOR) with 95% confidence intervals (CI) were reported. The comparison of the excluded and included participants for this study was analyzed for homogeneity using multiple tests, including the Kruskal-Wallis' test, Bartlett's test, and Chi-square test. Categorical variables with three or more subgroups and a significant distribution variation of p<0.05 were further analyzed using a two-sample Z test of proportions. All analyses were conducted using the Stata Statistical Software Release 16 (STATA Corp. 2019, College Station, TX, USA).

## Results

### Socio-demographics and sexual risk behaviors

The Lagos site of the TRUST/RV368 study enrolled 672 participants, of whom 478 engaged in the warts' treatment service, had complete behavioral data, and were included in these analyses. The median (interquartile range) age of the included participants was 22 (20–26) years; 272 (52%) had their sexual debut before 18 years of age; 30 (6%) were married, 135 (27%) were students, and 138 (27%) were employed (Table 1). More than half, 257 (56%) had engaged in condomless insertive or receptive anal sex with a male partner in the past 12 months, 347 (79%) were cisgender men, 279 (54%) had engaged in transactional sex, and 448 (90%) had multiple male sex partners in the past 12 months with a median of 6 (interquartile range: 4–12) male sex partners, and 2 (1–4) female sex partners. Ninety-two percent used lubricants during anal sex, and 342 (66%) were living with HIV, of which 219 (72%) were virally unsuppressed and 30 (11%) had a CD4 <200 cell/mm$^3$.

The 478 participants who were included in these analyses had similar characteristics to the 194 participants excluded (except for condomless insertive sex and not living with HIV) due to incomplete behavioral data or non-engagement with the warts management service (S1 Table). Excluded participants were more likely to report condomless insertive sex only (7.8% vs. 14.0%, p = 0.01), not living with HIV (43.3% vs. 28.5%, p<0.01), or having an unknown HIV status (24.7% vs. 0, p<0.01).

### Anogenital warts and sexual risk behaviors

The prevalence of anogenital, penile, and anal warts was 54%, 8%, and 43%, respectively, including 26 (5%) participants with both anal and penile warts (Table 1). Anogenital warts prevalence was higher among participants with age at sexual debut <18 years than among those at older age (57% vs. 34%; p = 0.005), and higher among transgender women than cisgender men (64% vs. 47%; p = 0.032). Furthermore, the prevalence of anogenital warts was higher among SGM who engaged solely in receptive anal sex compared to those who engaged solely in insertive sex (61% vs. 23%; p = 0.003), higher among SGM with multiple male sexual

**Table 1. Socio-demographics and sexual risk behaviors of anogenital warts among MSM.**

| Characteristics N = 478 | N [%] | Present n(%) | Absent n(%) | p-value |
|---|---|---|---|---|
| **Age (years)** | | | | |
| 18–24 | 315 [66.3] | 168 (45.1) | 147 (54.9) | 0.798 |
| ≥25 | 163 [33.7] | 84 (47.6) | 79 (52.4) | |
| **Age at sexual debut (years)** | | | | |
| 10–17 | 272 [51.6] | 147 (56.9) | 125 (43.1) | **0.005** |
| 18–26 | 206 [48.4] | 105 (34.4) | 101 (65.6) | |
| **Marital status** | | | | |
| Single | 434 [92.5] | 231 (46.1) | 203 (53.1) | 0.888 |
| Married/cohabiting | 30 [6.0] | 16 (45.7) | 14 (54.3) | |
| Separated/divorced/widowed | 14 [1.5] | 5 (36.6) | 9 (63.4) | |
| **Educational level** | | | | |
| ≤Secondary | 340 [75.5] | 178 (41.8) | 162 (58.2) | 0.059 |
| Tertiary | 138 [24.5] | 74 (58.6) | 64 (41.4) | |
| **Currently a student** | | | | |
| No | 343 [72.8] | 167 (42.9) | 176 (57.1) | 0.195 |
| Yes | 135 [27.2] | 85 (54.0) | 50 (46.0) | |
| **Occupational status** | | | | |
| Unemployed | 340 [73.0] | 190 (48.4) | 150 (51.6) | 0.313 |
| Employed | 138 [27.0] | 62 (39.4) | 76 (60.6) | |
| **Gender identity** | | | | |
| Cisgender man | 374 [79.2] | 196 (47.2) | 178 (52.8) | **0.032** |
| Transgender woman | 71 [10.0] | 40 (63.5) | 31 (36.5) | |
| Non-binary | 33 [10.8] | 16 (20.7) | 17 (79.3) | |
| **Sexual orientation** | | | | |
| Homosexual | 205 [40.5] | 112 (50.5) | 93 (49.5) | 0.385 |
| Bisexual | 272 [59.4] | 139 (42.8) | 133 (57.2) | |
| Queer | 1 [0.1] | 1 (100.0) | 0 (0.0) | |
| **Sexual position with male partners** | | | | |
| Engaged in insertive sex only | 75 [21.0] | 28 (23.4) | 47 (76.6) | **0.003** |
| Engaged in insertive and receptive sex | 268 [56.3] | 144 (48.2) | 124 (51.8) | |
| Engaged in receptive sex only | 135 [22.7] | 80 (61.3) | 55 (38.7) | |
| **Sexual position with female partners** | | | | |
| No insertive sex | 291 [45.5] | 163 (53.7) | 128 (46.3) | 0.085 |
| Engaged in insertive sex (vaginal or anal) | 187 [54.5] | 89 (39.5) | 98 (60.5) | |
| **Condomless sex with male sexual partners in past 12 month** | | | | |
| Always use condoms | 221 [44.0] | 106 (42.7) | 115 (57.3) | 0.262 |
| Condomless during receptive sex only | 110 [23.8] | 74 (55.1) | 36 (44.9) | |
| Condomless during insertive sex only | 37 [11.1] | 17 (26.3) | 20 (73.7) | |
| Condomless during both insertive and receptive sex | 110 [21.1] | 55 (52.8) | 55 (47.2) | |
| #**Condomless sex with female sexual partners in past 12 month** | | | | |
| No | 107 [39.2] | 53 (48.2) | 54 (51.8) | 0.241 |
| Yes | 80 [60.8] | 36 (33.9) | 44 (66.1) | |
| **Multiple male sexual partners in past 12 months** | | | | |
| No | 30 [10.1] | 14 (16.9) | 16 (83.1) | **0.004** |
| Yes | 448 [89.9] | 238 (49.2) | 210 (50.8) | |
| # **Multiple female sexual partners in past 12 months** | | | | |
| No | 55 [30.3] | 27 (39.5) | 28 (60.5) | 0.997 |

(*Continued*)

**Table 1.** (Continued)

| Characteristics N = 478 | N [%] | Present n(%) | Absent n(%) | p-value |
|---|---|---|---|---|
| Yes | 132 [69.7] | 62 (39.5) | 70 (60.5) | |
| **Transactional sex in past 12 months** | | | | |
| None | 199 [46.1] | 109 (45.0) | 90 (55.0) | 0.548 |
| Received money/gifts only | 163 [35.1] | 82 (43.1) | 81 (56.9) | |
| Paid money/gifts only | 47 [5.8] | 26 (66.7) | 21 (33.3) | |
| Both paid & received money/gifts | 69 [13.0] | 35 (47.5) | 34 (52.5) | |
| **Lubricant use during sex** | | | | |
| No | 31 [7.9] | 17 (28.6) | 14 (71.4) | 0.171 |
| Yes | 447 [92.1] | 235 (47.4) | 212 (52.6) | |
| **Easy access to lubricants** | | | | |
| No | 195 [38.6] | 107 (50.2) | 88 (49.8) | 0.428 |
| Yes | 283 [61.4] | 145 (43.3) | 138 (56.7) | |
| **HIV status** | | | | |
| Without HIV | 136 [34.2] | 45 (26.8) | 91 (73.2) | **0.001** |
| Living with HIV | 342 [65.8] | 207 (55.9) | 135 (44.1) | |
| *ART | | | | |
| No | 235 [74.9] | 144 (51.8) | 109 (48.2) | 0.162 |
| Yes | 89 [25.1] | 63 (68.0) | 26 (32.0) | |
| *Viral load | | | | |
| ≤1000 copies/mL | 123 [28.3] | 81 (58.8) | 42 (41.2) | 0.690 |
| >1000 copies/mL | 219 [71.7] | 126 (54.7) | 93 (45.3) | |
| *CD4 count | | | | |
| ≥200 cells/mm$^3$ | 312 [89.3] | 190 (56.2) | 122 (43.8) | 0.879 |
| <200 cells/mm$^3$ | 30 [10.7] | 17 (53.6) | 13 (46.4) | |
| **Recruitment waves** | | | | |
| 0–4 | 110 [26.5] | 61 (59.0) | 49 (41.0) | 0.204 |
| 5–9 | 142 [26.4] | 80 (42.9) | 62 (57.1) | |
| 10–14 | 99 [17.6] | 44 (32.9) | 55 (67.1) | |
| 15–19 | 86 [21.3] | 42 (42.0) | 44 (58.0) | |
| 20–24 | 38 [7.3] | 22 (46.6) | 16 (53.4) | |
| 25–29 | 3 [0.9] | 3 (100.0) | 0 (0.0) | |
| **Warts present** | | | | |
| Anogenital warts [yes] | 252 [54.0] | | | |
| Penile warts [yes] | 44 [7.5] | | | |
| Anal warts [yes] | 234 [43.2] | | | |
| Anal & penile warts [yes] | 26 [4.7] | | | |

*computed for Persons living with HIV group

#computed for those who had female sex partner; [%]–weighted column percentage (%)–weighted row percentages; All p-values in the table were calculated using weighted Pearson's chi-squared test and p<0.05 is considered statistically significant

partners than with a single partner (49% vs. 17%; p = 0.004), and higher among SGM living with HIV than those without HIV (56% vs. 27%; p = 0.001). Further exploratory bivariable analyses of our data did not show associations between anogenital warts and HIV (S2 Table), condom use, ART use, and HIV viral suppression (S3 Table). While the prevalence of anal warts was higher among SGM who reported their sexual debut as 10–17 years compared to

**Table 2. Anal and penile warts sexual risk behaviors by key socio-demographics and sexual risk behaviors variables.**

| Characteristics | Anal warts (N = 234/478) | | | Penile warts (N = 44/478) | | |
|---|---|---|---|---|---|---|
| | Present n (%) | Absent n (%) | p-value | Present n (%) | Absent n (%) | p-value |
| **Age (years)** | | | | | | |
| 18–24 | 160 (43.1) | 155 (56.9) | 0.976 | 23 (6.3) | 292 (93.7) | 0.339 |
| ≥ 25 | 74 (43.4) | 89 (56.6) | | 21 (9.7) | 142 (90.3) | |
| **Age at sexual debut (years)** | | | | | | |
| 10–17 | 138 (55.4) | 134 (44.6) | **0.001** | 23 (6.3) | 249 (93.7) | 0.482 |
| 18–26 | 96 (30.3) | 110 (69.7) | | 21 (8.6) | 185 (91.4) | |
| **Currently a student** | | | | | | |
| No | 155 (39.7) | 188 (60.3) | 0.127 | 31 (7.5) | 312 (92.5) | 0.993 |
| Yes | 79 (52.6) | 56 (47.4) | | 13 (7.4) | 122 (92.6) | |
| **Occupational status** | | | | | | |
| Unemployed | 178 (46.7) | 162 (53.3) | 0.148 | 34 (7.5) | 306 (92.5) | 0.940 |
| Employed | 56 (34.0) | 82 (66.0) | | 10 (7.2) | 128 (92.8) | |
| **Gender identity** | | | | | | |
| Cisgender men | 180 (44.7) | 194 (55.3) | **0.014** | 38 (8.2) | 336 (91.8) | 0.652 |
| Transgender women | 39 (60.3) | 32 (39.7) | | 4 (5.1) | 67 (94.9) | |
| Non-binary | 15 (16.6) | 18 (83.4) | | 2 (4.2) | 31 (95.8) | |
| **Sexual orientation** | | | | | | |
| Homosexual | 105 (47.1) | 100 (52.9) | 0.447 | 15 (6.5) | 190 (93.5) | 0.659 |
| Bisexual | 128 (40.6) | 144 (59.4) | | 29 (8.1) | 243 (91.9) | |
| Queer | 1 (100.0) | 0 (0.0) | | 0 (0.0) | 1 (100.0) | |
| **Sexual position with male partners** | | | | | | |
| Engaged in insertive sex only | 20 (19.5) | 55 (80.5) | **0.001** | 13 (10.1) | 62 (89.9) | 0.317 |
| Engaged in insertive and receptive sex | 137 (45.5) | 131 (54.5) | | 19 (5.2) | 249 (94.8) | |
| Engaged in receptive sex only | 77 (59.5) | 58 (40.5) | | 12 (10.6) | 123 (89.4) | |
| **Multiple male sexual partners in past 12 months** | | | | | | |
| No | 14 (16.9) | 16 (83.1) | **0.008** | 1 (0.5) | 29 (99.5) | **<0.001** |
| Yes | 220 (46.2) | 228 (53.8) | | 43 (8.2) | 405 (91.8) | |
| **Lubricant use during sex** | | | | | | |
| No | 17 (28.6) | 14 (71.4) | 0.245 | 2 (2.6) | 29 (97.4) | 0.135 |
| Yes | 217 (44.5) | 230 (55.5) | | 42 (7.9) | 405 (92.1) | |

All p values in the table were calculated using Fisher's exact test, and statistical significance was set at p<0.05. n(%) = frequency and weighted row percentage.

those who reported 18 years and older 55% vs. 30%; p = 0.001); identified as transgender women compared to cisgender men (60% vs. 45%; p = 0.014); engaged in receptive anal sex only with male partners than among those who engaged exclusively in insertive anal sex (60% vs. 20%; p<0.001); and reported multiple male sex partners in the last 12 months (46% vs. 17%; p = 0.008), the prevalence of penile warts was higher only among those who reported multiple male sexual partners in the last 12 months (8% vs. 1%; p<0.001) as shown in Table 2.

## Correlates of anogenital warts

Table 3 outlines the factors independently associated with anogenital warts among Nigerian SGM. The odds of being diagnosed with anogenital warts were almost three times higher (AOR: 2.97; 95% CI: 1.44–6.14) among SGM living with HIV than among those without HIV. Compared to SGM who engaged only in insertive anal sex, those who engaged only in

**Table 3. Multivariable analysis showing factors associated with anogenital warts among MSM in Nigeria.**

| Factors | n | Crude OR (95% CI) | p value | Adjusted OR (95% CI) | p value |
|---|---|---|---|---|---|
| **Age at sexual debut (years)** | | | | | |
| 10–17 | 315 | ref. | | ref. | |
| 18–26 | 163 | 0.40 (0.21–0.76) | 0.006 | **0.53 (0.28–0.98)** | **0.046** |
| **Educational level** | | | | | |
| ≤Secondary | 340 | ref. | | ref. | |
| Tertiary | 138 | 1.97 (0.97–4.01) | 0.062 | 1.78 (0.90–3.51) | 0.096 |
| **Gender Identity** | | | | | |
| Cisgender men | 374 | ref. | | ref. | |
| Transgender women | 71 | 1.95 (0.84–4.51) | 0.120 | 0.96 (0.37–2.48) | 0.944 |
| Non-binary | 33 | 0.29 (0.08–1.11) | 0.071 | **0.20 (0.05–0.71)** | **0.014** |
| **Sexual position with male partners** | | | | | |
| Engaged in insertive sex only | 75 | ref. | | ref. | |
| Engaged in both insertive and receptive sex | 268 | **5.17 (2.02–13.22)** | **0.001** | 1.88 (0.79–4.47) | 0.153 |
| Engaged in receptive only | 135 | **3.04 (1.24–7.43)** | **0.015** | **3.49 (1.25–9.75)** | **0.017** |
| **Sexual position with female partners** | | | | | |
| No insertive sex | 291 | ref. | | * | |
| Engaged in insertive sex (vaginal or anal) | 187 | 0.56 (0.29–1.09) | 0.086 | - | - |
| **Multiple male sexual partners in past 12 months** | | | | | |
| No | 30 | ref. | | ref. | |
| Yes | 448 | 4.76 (1.56–14.86) | 0.007 | **7.26 (2.11–24.87)** | **0.002** |
| **HIV status** | | | | | |
| Without HIV | 136 | ref. | | ref. | |
| Living with HIV | 342 | **3.46 (1.63–7.34)** | **0.001** | **2.97 (1.44–6.14)** | **0.003** |

OR–Odds ratio ref.–referent group; CI–confidence interval; Statistical significance is p-value <0.05; *—variable eliminated from the multivariable model

receptive anal sex had three times higher odds of having anogenital warts (AOR: 3.49; 95%CI: 1.25–9.75). Furthermore, SGM who had multiple male sexual partners had seven times higher odds of being diagnosed with anogenital warts than those with one partner (AOR: 7.26; 95% CI: 2.11–24.87). The odds of anogenital warts were 47% lower among those who reported their sexual debut as 18 and older (AOR: 0.53; 95%CI: 0.28–0.98) compared to those under 18 years. Those who identified as non-binary men were 80% less likely to be diagnosed with anogenital warts compared to cisgender men (AOR: 0.20; 95%CI: 0.05–0.71).

## Discussion

This is one of the few studies to characterize and identify the correlates of anogenital warts in SGM in sub-Saharan Africa. Our findings suggest that the prevalence of anogenital warts was high among predominantly young SGM, with a median age of 22 years. In addition, the 54% prevalence of warts at the Lagos site was higher than the 12% reported from an early evaluation of anogenital warts at the Abuja site of TRUST/RV368 [15]. In Abuja, the clinic staff may have underdiagnosed warts during the physical examination and the SGM included in the analytic sample were older with a lower prevalence of HIV, all of which may have contributed to the study site differences.

Anal warts were substantially more common than penile warts, paralleling the results reported among an older and more sexually experienced cohort of MSM in Australia (anal warts: 20% vs. genital warts: 9%) [22]. Higher prevalence and incidence of warts and HPV

have also been reported in the anal canal than in other genital sites among MSM in Denmark [23], and Australia [24]. This may be attributed to the higher susceptibility of the anal mucosa during anal sex as compared to the keratinized epithelium of the penis.

Early sexual debut is a strong predictor of number of lifetime sexual partners and STI [25]. This was corroborated in our study, as the prevalence of anogenital warts was fifty-three percent lower among SGM who delayed initiation of sex until after 18 years compared to those with an earlier age of initiation. Additionally, prevalence of anogenital warts was 7-fold higher among SGM who reported multiple male sexual partners compared to their peers who did not. Young SGM in Nigeria who initiate sex early are more likely to engage in riskier sexual and social behaviors with limited access to prevention interventions and higher exposure to LR- and HR-HPV, resulting in anogenital warts or penile intraepithelial neoplasia.(27) Behavior change interventions that focus on this age group could consider being holistic in their coverage of STIs to also include anogenital warts. Globally, persons whose gender identities do not match with their sex assigned at birth including transgender, non-binary and nonconforming sub-populations are disproportionately affected by human immunodeficiency virus (HIV) and other STIs [26, 27]. Scientific data on sexual, social behaviors and health outcomes including STIs among transgender and non-conforming persons are scarce compared to cisgender men in Nigeria. Our study showed that although the prevalence of anogenital warts was highest among transgender women and lowest among non-binary men compared to cisgender men, non-binary men were less likely to be diagnosed with anogenital warts than their cisgender male peers. It is possible that non-binary men in Nigeria are less exposed to stigma and discrimination, and they have better access to prevention and health services compared to transgender women and cisgender men. More studies are needed to understand the epidemiology of anogenital warts and other STIs among the different typologies of sexual and gender minorities to better inform more gender affirmative services are needed in Nigeria.

SGM living with HIV in our study were more than three times likely to be diagnosed with anogenital warts than their peers who are not living with HIV. Notably, it is not uncommon for SGM living with HIV to have HPV coinfections. Machalek et al (2012) in a meta-analysis reported prevalence of 93% of any HPV among MSM living with HIV compared to those not living with HIV [8]. Moreover, Nowak et al. (2016) reported a higher prevalence of wart-associated HPV6 (43% vs. 20%) and HPV11 (28% vs. 13%) among SGM living with and without HIV, respectively, from the Abuja site of our TRUST/RV368 cohort study [15]. HIV may further exacerbate the presentation of anogenital warts, as it is reported to be associated with the reactivation and lower immune control of latent HPV infection [28]. Interestingly, prevalence of anal warts in our study of relatively young SGM was higher among those on ART than those not on treatment probably because of shorter duration on ART, and incorrect or irregular use of ART. This may not be unconnected to the pervasive stigma, discrimination and homophobic attitudes of healthcare providers in Nigeria that may limit access of SGM to prevention, treatments, and care services in a timely manner. Furthermore, our study did not show statistically significant associations between anogenital warts and high HIV plasma viral load or lower CD4+ counts. Our study may have been under-powered to evaluate HIV-related immunosuppression and prevalence of anogenital warts, as only 9% of the sample reported a CD4 count of <200 cell/mm$^3$. Using a single CD4 measurement that is prone to internal and external influences, may result in misclassification of immune status. CD4 trajectories from longitudinal data are recommended as better predictors of incident anogenital warts among persons living with HIV.(29) Lastly, the lack of association between anogenital warts and viral load could be due to the stage of infection with many participants being newly diagnosed and are yet to experience chronic HIV.

Receptive anal sex was associated with increased odds of anogenital warts. Although 80% of the study participants reported lubricant use and almost two-thirds reported easy access to

water-based lubricants, the lubricants were not protective against anogenital warts in this study. The etiology of HPV inducing lesions is not well understood, however, pathogenesis of warts is reported to be associated with abrasions and scars resulting from trauma (Koebner phenomenon) which may explain why non-use of lubricants during anal intercourse heightens recurrence of abrasions and development of anogenital warts [29]. Jin et al. (2007) reported an association between non-penetrative sexual practices, such as fingering and fisting that increase laceration and scarring, with anal warts [22]. Health interventions for SGM in Nigeria that increase access to and promote the use of condom and compatible water-based lubricants may maintain the integrity of the mucosal epithelial barrier.

This study had some limitations. The RDS recruitment technique could have introduced some bias in the way participants were recruited into the study as it relied on the networks of peers and not a random sample. To account for this, RDS weights were used in our analyses to adjust for the individual social network sizes. Information on the number of lifetime sexual partners, a strong predictor of anogenital warts, was not collected in the behavioral questionnaire. Information on sexual behavior was collected through self-report for a period of 12 months prior to study enrollment, which may be vulnerable to recall, social desirability, and other biases. Lastly, it is possible that the higher prevalence of anogenital warts reported in this study was because those selected for these specific analyses were more likely to participate in the warts management services.

## Conclusion

The prevalence of anal warts among SGM in Nigeria seems high, particularly among those living with HIV. Offering warts management services as an inclusive component of comprehensive HIV prevention, treatment, and care services may alleviate the high physical, psychosocial and economic burden of anogenital warts. This will go a long way in improving the quality of life of SGM. Furthermore, targeted HPV vaccination of SGM boys before their sexual debut as well as catch-up vaccinations in SGM-friendly clinics would be a cost-effective strategy with significant impact.

## Supporting information

**S1 Table. Sensitivity comparison of sociodemographic and sexual behaviour factors between SGM included and excluded from the analyses.**
(DOCX)

**S2 Table. Bivariate analysis of condom-use and HIV status among SGM.**
(DOCX)

**S3 Table. Binary logistic regression of selected factors associated with anogenital warts among SGM living with HIV.**
(DOCX)

## Acknowledgments

We would like to thank all study participants and the RV368 study team. The TRUST/RV368 Study Group includes Principal Investigators: Manhattan Charurat (IHV, University of Maryland, Baltimore, MD, USA), Julie Ake (MHRP, Walter Reed Army Institute of Research, Silver Spring, MD, USA); Co-Investigators: Sylvia Adebajo, Stefan Baral, Erik Billings, Trevor Crowell, George Eluwa, Charlotte Gaydos, Afoke Kokogho, Hongjie Liu, Jennifer Malia, Olumide Makanjuola, Nelson Michael, Nicaise Ndembi, Jean Njab, Rebecca Nowak, Oluwasolape

Olawore, Zahra Parker, Sheila Peel, Habib Ramadhani, Merlin Robb, Cristina Rodriguez-Hart, Eric Sanders-Buell, Sodsai Tovanabutra; Institutions: Institute of Human Virology at the University of Maryland School of Medicine (IHV-UMB), University of Maryland School of Public Health (UMD SPH), Johns Hopkins Bloomberg School of Public Health (JHSPH), Johns Hopkins University School of Medicine (JHUSOM), U.S. Military HIV Research Program (MHRP), Walter Reed Army Institute of Research (WRAIR), Henry M. Jackson Foundation for the Advancement of Military Medicine (HJF), Henry M. Jackson Foundation Medical Research International (HJFMRI), Institute of Human Virology Nigeria (IHVN), International Centre for Advocacy for the Right to Health (ICARH), The Initiative for Equal Rights (TIERS), Population Council Nigeria, Imperial College London.

## Author Contributions

**Conceptualization:** Sylvia B. Adebajo, Rebecca G. Nowak, Charles Ekeh, Stefan D. Baral, Manhattan E. Charurat, Trevor A. Crowell.

**Data curation:** Rebecca G. Nowak, Ruxton Adebiyi, Habib O. Ramadhani, Trevor A. Crowell.

**Formal analysis:** Ruxton Adebiyi, Habib O. Ramadhani.

**Funding acquisition:** Sylvia B. Adebajo, Manhattan E. Charurat, Trevor A. Crowell.

**Investigation:** Rebecca G. Nowak, Elizabeth Shoyemi, Charlotte A. Gaydos, Manhattan E. Charurat, Trevor A. Crowell.

**Methodology:** Sylvia B. Adebajo, Rebecca G. Nowak, Ruxton Adebiyi, Elizabeth Shoyemi, Habib O. Ramadhani, Charlotte A. Gaydos, Stefan D. Baral.

**Project administration:** Sylvia B. Adebajo, Elizabeth Shoyemi, Charles Ekeh.

**Resources:** Manhattan E. Charurat.

**Software:** Ruxton Adebiyi.

**Supervision:** Sylvia B. Adebajo, Rebecca G. Nowak, Charlotte A. Gaydos, Julie A. Ake, Manhattan E. Charurat, Trevor A. Crowell.

**Writing – original draft:** Sylvia B. Adebajo, Elizabeth Shoyemi.

**Writing – review & editing:** Sylvia B. Adebajo, Rebecca G. Nowak, Habib O. Ramadhani, Charlotte A. Gaydos, Julie A. Ake, Stefan D. Baral, Manhattan E. Charurat, Trevor A. Crowell.

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
