## [Decision Letter · Decision Letter 0]

9 Jul 2022

PGPH-D-22-00848

Prevalence and factors associated with anogenital warts among sexual and gender minorities attending a trusted community health Center in Lagos, Nigeria

Dear Dr. Adebajo,

Thank you for submitting your manuscript to PLOS Global Public Health. After careful consideration, we feel that it has merit but does not fully meet PLOS Global Public Health’s publication criteria as it currently stands. Therefore, we invite you to submit a revised version of the manuscript that addresses the points raised during the review process.

Overall, the reviewers had relatively favorable comments. Please thoughtfully consider their comments, and provide an item-by-item response to every comment. If you make an edit, please paste your edit in the response to reviewers document.

We look forward to receiving your revised manuscript.

Kind regards,

Abram L. Wagner, PhD, MPH

Academic Editor

Journal Requirements:

1.  Please amend your detailed online Financial Disclosure statement. This is published with the article. It must therefore be completed in full sentences and contain the exact wording you wish to be published.

State what role the funders took in the study. If the funders had no role in your study, please state: “The funders had no role in study design, data collection and analysis, decision to publish, or preparation of the manuscript.”

2. Please update your online Competing Interests statement. If you have no competing interests to declare, please state: “The authors have declared that no competing interests exist.”

3. We do not publish any copyright or trademark symbols that usually accompany proprietary names, eg (R), (C), or TM  (e.g. next to drug or reagent names). Please remove all instances of trademark/copyright symbols throughout the text, including ® (Determine®, Uni-gold®) on page 5.

Additional Editor Comments (if provided):

Reviewers' comments:

Reviewer's Responses to Questions

**Comments to the Author**

1. Does this manuscript meet PLOS Global Public Health’s publication criteria? Is the manuscript technically sound, and do the data support the conclusions? The manuscript must describe methodologically and ethically rigorous research with conclusions that are appropriately drawn based on the data presented.

Reviewer #1: Yes

Reviewer #2: Yes

2. Has the statistical analysis been performed appropriately and rigorously?

Reviewer #1: I don't know

Reviewer #2: I don't know

3. Have the authors made all data underlying the findings in their manuscript fully available (please refer to the Data Availability Statement at the start of the manuscript PDF file)?

Reviewer #1: Yes

Reviewer #2: Yes

4. Is the manuscript presented in an intelligible fashion and written in standard English?

Reviewer #1: Yes

Reviewer #2: Yes

5. Review Comments to the Author

Reviewer #1: Overall this is a well crafted study of the determinants of anogenital warts among men who have sex with men in Lagos, Nigeria. I have two minor comments regarding interpretation of results: (1) In discussing the higher prevalence of anogential warts among men living with HIV, the authors remark that “HIV may further exacerbate the presentation of [216] anogenital warts, as it has been associated with reactivation and lower immune control of latent [217] HPV infection.(28),” but was there more (condomless) receptive anal sex among HIV-positive study participants that may also explain the relationship with rates of anogential warts? (2) There is passing reference to the existence of HPV vaccines. What access (or barriers to access) are there to vaccination in the study area?

Reviewer #2: Dear authors,

You have chosen a very important topic that is seldom addressed in public health. I am positive that your article will contribute to efforts of raising awareness to the high prevalence of anogenital warts particularly among SMG in LMIC and those who live with HIV.

I fully support its publications, with minor revisions and clarifications to the comments below.

Methods/Statistical analysis:

• The data was collected using RDS, but all percentages are presented without adjusting for bias in sample collection method. Please explain and include information on potential bias due to the sampling method. Also, mention if the differences found between adjusted and unadjusted prevalence were large.

• The definition of the variable “occupational status” is not clear. How students who worked were classified?

• Some independent variables are likely correlated with one another such as age and being a student; and ART, viral load and CD4 count. It is not clear how confounders were managed in the multivariate analysis.

• The AOR for those on ART vs. not on ART were not presented. Any reason for that?

Discussion

• The lack of association between anogenital warts and viral load in the study warrant some explanation, as it does not seem to be related to the sample size as for CD4 count. Additionally, suggest adding an explanation for the counterintuitive result of those on ART to have higher prevalence of warts.

• Several results can lead to recommendations for programme managers, as the excellent one on wart management for PLHIV mentioned in the conclusion. Regarding other recommendations made, specific suggestions are:

a. Behavioral risk interventions: It is not clear what is being recommended regarding behavioral risk interventions that could address anogenital warts beyond what is already addressed (i.e, condom and lubricant use). Suggest deleting this recommendation by deleting the sentence on lines 207-208.

b. Condom promotion and access. Good point. On line 235 add “and” (…use of condom and compatible water-based lubricants). Add in the conclusion.

c. Vaccination of boys. I am not convinced that this would be the most effective strategy as vaccination coverage at populational level would have to be very high to reach SMG before sexual debut. Suggest focusing on catch-up targeted vaccination for SMG in friendly clinics instead. The rational is the likelihood of co-infection with multiple viral HPV genotypes and the potential cross-protection effect when quadrivalent or nonavalent vaccines are used (see US CDC recommendations for vaccinating all adults).

d. Long-term sequelae (line 248-249) goes beyond the scope of the paper. The sentence could be deleted and replaced by a adding a sentence the need to address anogenital warts to improve quality of life, including sexual life.

Minor points:

• In the introduction, besides mentioning the STI-related stigma and discrimination, suggest mentioning the “impact” of anogenital on quality of life. This is best described among women, but some small studies have been addressing this important health consequence.

• The list of independent variables is not complete. Suggest either explaining why some were not mentioned or refer to the complete list on table.

• Line 154, rephrase “The 478 participants who were included in these analyses had similar characteristics to the 194 155 participants who were excluded…” to “The 478 participants who were included in these analyses had similar characteristics to the 194 155 participants who were excluded, with the exception of condomless insertive sex only and…..

• In the conclusion, suggest changing “The prevalence of anal warts among SGM in Nigeria is high..” to “seems high”, considering the inherited bias of the study addressed in the limitations.

Congratulations for the article.

6. PLOS authors have the option to publish the peer review history of their article (what does this mean?). If published, this will include your full peer review and any attached files.

**Do you want your identity to be public for this peer review?** For information about this choice, including consent withdrawal, please see our Privacy Policy.

Reviewer #1: **Yes: **Barry Adam

Reviewer #2: No

---

## [Decision Letter · Decision Letter 1]

4 Oct 2022

Prevalence and factors associated with anogenital warts among sexual and gender minorities attending a trusted community health Center in Lagos, Nigeria

PGPH-D-22-00848R1

Dear Dr Adebajo,

We are pleased to inform you that your manuscript 'Prevalence and factors associated with anogenital warts among sexual and gender minorities attending a trusted community health Center in Lagos, Nigeria' has been provisionally accepted for publication in PLOS Global Public Health.

Best regards,

Julia Robinson

Executive Editor

Reviewer Comments (if any, and for reference):

Reviewer's Responses to Questions

**Comments to the Author**

1. If the authors have adequately addressed your comments raised in a previous round of review and you feel that this manuscript is now acceptable for publication, you may indicate that here to bypass the “Comments to the Author” section, enter your conflict of interest statement in the “Confidential to Editor” section, and submit your "Accept" recommendation.

Reviewer #2: All comments have been addressed

2. Does this manuscript meet PLOS Global Public Health’s publication criteria? Is the manuscript technically sound, and do the data support the conclusions? The manuscript must describe methodologically and ethically rigorous research with conclusions that are appropriately drawn based on the data presented.

Reviewer #2: Yes

3. Has the statistical analysis been performed appropriately and rigorously?

Reviewer #2: Yes

4. Have the authors made all data underlying the findings in their manuscript fully available (please refer to the Data Availability Statement at the start of the manuscript PDF file)?

Reviewer #2: Yes

5. Is the manuscript presented in an intelligible fashion and written in standard English?

Reviewer #2: Yes

6. Review Comments to the Author

Reviewer #2: I do not have any additional comments. Many thanks for addressing my concerns and suggestions and congratulations for contributing to this important topic.

7. PLOS authors have the option to publish the peer review history of their article (what does this mean?). If published, this will include your full peer review and any attached files.

**Do you want your identity to be public for this peer review?** For information about this choice, including consent withdrawal, please see our Privacy Policy.

Reviewer #2: **Yes: **Maeve Brito de Mello
